# Alternative Targets to Fight Alzheimer’s Disease: Focus on Astrocytes

**DOI:** 10.3390/biom11040600

**Published:** 2021-04-19

**Authors:** Marta Valenza, Roberta Facchinetti, Giorgia Menegoni, Luca Steardo, Caterina Scuderi

**Affiliations:** 1Department of Physiology and Pharmacology “Vittorio Erspamer”, SAPIENZA University of Rome—P.le A. Moro, 5, 00185 Rome, Italy; martavalenza@gmail.com (M.V.); roberta.facchinetti@uniroma1.it (R.F.); giorgiamenegoni95@gmail.com (G.M.); luca.steardo@uniroma1.it (L.S.); 2Università Telematica Giustino Fortunato—Via Raffaele Delcogliano, 12, 82100 Benevento, Italy

**Keywords:** Alzheimer’s disease, astrocytes, astrogliosis, beta amyloid, neuroinflammation, neuroprotection, reactive gliosis, palmitoylethanolamide

## Abstract

The available treatments for patients affected by Alzheimer’s disease (AD) are not curative. Numerous clinical trials have failed during the past decades. Therefore, scientists need to explore new avenues to tackle this disease. In the present review, we briefly summarize the pathological mechanisms of AD known so far, based on which different therapeutic tools have been designed. Then, we focus on a specific approach that is targeting astrocytes. Indeed, these non-neuronal brain cells respond to any insult, injury, or disease of the brain, including AD. The study of astrocytes is complicated by the fact that they exert a plethora of homeostatic functions, and their disease-induced changes could be context-, time-, and disease specific. However, this complex but fervent area of research has produced a large amount of data targeting different astrocytic functions using pharmacological approaches. Here, we review the most recent literature findings that have been published in the last five years to stimulate new hypotheses and ideas to work on, highlighting the peculiar ability of palmitoylethanolamide to modulate astrocytes according to their morpho-functional state, which ultimately suggests a possible potential disease-modifying therapeutic approach for AD.

## 1. Introduction

Aducanumab, a monoclonal antibody directed against the aggregated form of the beta-amyloid peptide (Aβ), was the last unfruitful attempt to treat Alzheimer’s disease (AD). At the beginning of November 2020, experts of the Peripheral and Central Nervous System Drugs Advisory Committee of the Food and Drug Administration expressed some concerns about the real efficacy of aducanumab, thus hindering its marketing claiming [1]. The AD field had high expectations for the aducanumab clinical trials, primarily because this human IgG1 monoclonal antibody was designed to selectively bind Aβ aggregates, including soluble oligomers and insoluble fibrils but not monomers [2], suggesting the possibility to overcome previously failed approaches of other anti-Aβ antibodies. Unfortunately, the aims were never met, despite they had been well demonstrated at the preclinical level and in the early stages of the clinical trial. This event reveals once again the limitations of both basic and medical research anxiously focused on counteracting Aβ in AD [3].

AD is the most common form of dementia in the elderly, affecting about 47 million people worldwide [4]. Most AD cases are sporadic, affecting people older than 65 years old, and aging represents the greatest risk factor [5]. As life expectancy increases, it is reasonable to foresee that the number of AD patients will grow in the next decades. However, other risk factors have been identified besides old age. Growing epidemiological data support the existence of a link between metabolic disorders and AD [6,7,8,9,10], and a correlation between head injury and future risk of dementia has also been suggested. The risk of developing AD or vascular dementia is increased in many pathological conditions of the heart and blood vessels, including heart failure, diabetes, stroke, high blood pressure, and high cholesterol level [11]. Family history and heredity are the most important risk factors for the genetic form of this disease, which affects about 1% of individuals with AD, and whose symptoms appear as early as 35 years old [12]. In contrast to heredity and aging, which are nonmodifiable factors, other risk factors could be controlled through general lifestyle improvement and effective management of unhealthy conditions. Indeed, healthy aging, which includes both physical and mental exercise, a balanced diet, staying socially active, and avoiding smoking, preserves both body and brain wellness and reduces the risk for developing dementia [13,14,15].

At a molecular level, the presence of two peculiar hallmarks characterizes the AD brain: (I) senile plaques, formed by the deposition of Aβ peptides in the extracellular space, and (II) neurofibrillary tangles (NFTs), due to the hyperphosphorylation of microtubule-associated tau proteins. A growing body of evidence indicates, however, that senile plaques and NFTs alone are not responsible for the cognitive impairments observed in AD [16]. Neuroinflammation and abnormal astrocytic and microglia responses exert a pivotal role in AD pathogenesis and progression, thus highlighting the complexity of this pathology [17,18,19].

AD can be considered as a continuum that spans decades [20], with brain modifications that begin 10–20 years before the clinical manifestations and change throughout the disease progression [21]. Various clinical stages have been classified, such as asymptomatic preclinical, prodromal, mild, moderate, and severe AD [22,23], also referred to as stage 1 to stage 6 [24,25]. Each stage is characterized by peculiar molecular changes that could represent possible targets for different therapeutic approaches [26,27,28].

AD is a neurodegenerative disease that impacts memory and cognition. In addition to the progressive impairment in mental abilities, other debilitating noncognitive symptoms usually appear, including sleep disturbances, loss of appetite, and neuropsychiatric conditions, including depression and/or apathy [29,30]. In the latest stages, symptoms worsen enough to interfere with daily activities such that people suffering from AD need continuous care. As a result, the economic burden of AD is impressive, mainly because currently approved drugs are not curative. Despite decades of intense research, no treatments are available to halt, slow, or cure AD, and the therapy still relies on cholinesterase inhibitors (donepezil, rivastigmine, and galantamine), and the N-methyl-D-aspartate (NMDA) antagonist memantine. Any of these drugs slightly help to manage behavioral symptoms, preserve mental skills, and slow down the disease progression. However, their effects are reversible and lessen over time due to the continued progression of the disease [31,32].

A final and confirmed diagnosis of AD can only be made through postmortem identification of histopathological hallmarks. Whenever a patient is suspected to have AD, he/she is already in a mild or moderate stage of the pathology, and substantial irreversible neuronal dysfunction and loss have already occurred. Nowadays, clinicians concur that intervening at the earliest stage of the disease could lead to a better outcome [22,33]. To do so, it will be necessary to identify biological markers allowing diagnosis in the asymptomatic (or, at most, prodromal) stage of the disease to recognize asymptomatic at-risk individuals and refer them to the use of disease-modifying drugs. This approach could be insidious and difficult to achieve since it falls into the field of preventive care. Despite the preclinical stage of AD could represent a temporal window in which it may be possible to reduce the incidence and progression of the disease [34], few preclinical data are available so far at this stage of the pathology [35]. To develop preventive therapeutic approaches for AD in the coming years, the key neurobiological mechanisms of AD need to be clarified.

In this review, we discuss the most recent findings on both old and new mechanisms implicated in AD, with a particular reference to the role played by glial cells. The brain homeostatic functions exerted by glia could represent a novel perspective in AD management, offering new strategies to treat this disease.

## 2. Old and New Pathophysiological Mechanisms in AD

### 2.1. The Amyloid Cascade Hypothesis

The correlation between Aβ deposition and dementia has extensively been studied during the past decades, and the amyloidogenic pathway has been widely investigated as a target for drug development [36]. Alois Alzheimer himself described the presence of plaques during the histological examination of his patient’s—named Augustine—brain [37,38]. Later after, such plaques were recognized to be protein deposits, mainly Aβ peptides [39,40]. Several forms of Aβ peptides have been found in AD brains [41,42]. Longitudinal PET studies demonstrated that proteins begin to deposit about two decades before first symptoms appear [43]; thus, plaque formation is a slow and prolonged process. Plaques accumulate extensively throughout the cortex, with the occipital and temporal lobes being the most affected [44].

Aβ peptides are generated by the cleavage of the type I transmembrane amyloid precursor protein (APP), a protein expressed ubiquitously, which biological functions remain unclear [45,46]. APP is particularly abundant in the brain, and evidence showed that it has trophic properties [47]. It plays a role in brain development by promoting neural stem cells (NSCs) proliferation, cell differentiation, and neuronal maturation [46,48]. APP seems necessary for synaptogenesis, synapse remodeling, and neurite outgrowth [49,50], as well as axonal outgrowth after injury in the adult brain [51]. A neuroregenerative role for brain APP has been hypothesized, even if the molecular mechanisms have not been elucidated yet. The production of APP increases in some physiological conditions, such as during neuronal maturation and differentiation, and in some pathological ones, including AD, brain trauma, and Down syndrome [52]. To complicate the picture, alternative transcriptional splicing could create 8 to 11 different APP isoforms [53].

The enzymatic processing of APP yields various peptides with distinct functions through three different proteolytic pathways, among which only one seems to be amyloidogenic. This process releases mainly two monomers of Aβ: about 90% is Aβ40, which is considered nontoxic because it does not self-aggregate much, and the remaining part is mainly constituted of longer Aβ peptides [54]. Being more hydrophobic and prone to aggregate than the shorter isoforms, the Aβ42 and Aβ43 could form oligomers and fibrils; thus, they are considered neurotoxic isoforms [36,55]. In addition to Aβ42 and Aβ43, some reports consider also the amyloid precursor protein intracellular domain to be involved in the pathophysiology of AD [52,56,57].

The nonamyloidogenic pathway is thought not to generate toxic Aβ and a third proteolytic pathway has been recently described, involving a η-secretase that cuts the APP extracellular domain releasing a soluble ectodomain. The biological functions of all peptides yielded through this newly described pathway are yet to be disclosed.

Although most of the research studies investigated the neurotoxicity of Aβ peptides, they also exert biological functions. They are not abundantly expressed, even in AD brains [58], and they execute trophic actions, including cell fate specification and proliferation. Exogenous application of soluble and fibrillary Aβ peptides (but not oligomeric forms) stimulates human embryonic stem cells (ESCs) proliferation [59]. Oligomeric Aβ peptides reduce the proliferative potential of human NSC, promoting their differentiation toward glial instead of neuronal cells [60]. The Aβ40 seems to preferably enhance neurogenesis, whereas the Aβ42 seems to promote gliogenesis [61,62]. Some authors have also observed neurogenesis induced by oligomers of Aβ42, and not Aβ40, in rat hippocampal NSCs [63]. Further studies are warranted since these contradictory results are probably due to the different forms of Aβ used.

The amyloid cascade hypothesis states that the progressive accumulation and oligomerization of Aβ42 creates diffuse plaques in the brain parenchyma, causing neuroinflammation and, later, neurofibrillary tangles, ultimately leading to synaptic dysfunction or loss, and neuronal death [36,64]. This hypothesis has been formulated after having identified the APP gene on chromosome 21, together with the observation that people affected by Down syndrome develop AD-like symptoms early in life. Several pathogenic coding mutations in the APP gene have been identified and linked to the onset of autosomal dominant AD [64,65,66]. This hypothesis is supported by the correlation between autosomal dominant mutations in both APP and genes coding for parts of the secretase, such as presenilin (PSEN) 1, PSEN2, with the incidence of AD [36,64,67].

However, reduced clearance of Aβ peptides could also account for their accumulation in the brain. A protein involved both in the clearance of Aβ and in its ability to aggregate and form fibrils is the apolipoprotein E (apoE) [68,69]. Homozygous carriers for the isoform ε4 have about a 12-fold higher risk to develop sporadic AD, while carriers of the less frequent ε2 isoform show a low risk for AD [70,71]. Despite all the findings that strongly support the amyloid cascade hypothesis, other data suggest instead that the accumulation of senile plaques in the brain does not correlate with cognitive impairment. Indeed, massive cerebral accumulation of Aβ plaques has also been observed in individuals without any cognitive impairment. Additionally, the reduction of Aβ load by immunotherapy does not improve cognition in AD patients [72]. Furthermore, all clinical trials carried out so far targeting either the production or the accumulation of Aβ have failed. The debate is fervent in the literature and undoubtfully more studies are needed to clarify the precise mechanism(s) by which Aβ deposits lead to tangle formation, and thus neurodegeneration [3].

### 2.2. Neurofibrillary Tangles

Neurofibrillary tangles are considered essential for the neuropathological diagnosis of AD [26]. They are intraneuronal bundles of filaments made of hyperphosphorylated microtubule-associated tau proteins [73]. Their accumulation causes a loss of cytoskeletal microtubules and tubulin-associated proteins, resulting in morphological modifications in neuronal dendrites and axons [74].

Since NFTs appearance in the brain seems to follow a pattern, in a seminal paper, Braak and Braak proposed to classify AD in six stages based on neuropathological findings [44,75].

At physiological conditions, the protein tau is mainly localized in the axon, and it is essential for the stabilization of microtubules [76]. Its phosphorylation is highly probable because tau has 85 potential sites of phosphorylation that are easily accessible because of the unfolded structure of the protein [77]. Tau has been found mislocalized (missorted) into the somatodendritic compartment at the early stages of AD. Since NFTs load correlates with cognitive decline and synapse loss [74], a role for tau missorting has been proposed in AD [78], which serves as diagnostic criteria and for the staging of disease progression [79]. Interestingly, abnormal phosphorylation of tau is detectable even before NFTs formation. In agreement, the reduction of tau has beneficial effects in preclinical AD models, whereas tau mislocalization from axons to dendrites has detrimental effects [80,81]. In general, the major modifications of tau found in AD are hyperphosphorylation, missorting, aggregation to oligomers and filaments forming paired helical filaments, dissociation from microtubules, and other post-translational modifications [78].

Mutations in genes encoding for tau have not been linked to AD. However, tau knockout mice show very mild neurite outgrowth changes and no microtubule-related defects [82,83]. Human findings showed that microtubule density is decreased in AD patients, but this reduction is surprisingly unrelated to tau abnormalities [84]. Consistent with the above, a simple loss of function of tau is not enough to explain the loss of microtubules observed in AD, and other mechanisms are probably involved.

Several tau-targeting therapies for AD have been proposed. These approaches are based mainly on (i) inhibition of kinases (responsible for aberrant tau phosphorylation), (ii) inhibition of tau aggregation, and (iii) stabilization of microtubules. Immunotherapies targeting tau in clinical trials have shown high toxicity and/or lack of efficacy and have been discontinued [85].

### 2.3. Unfolded Protein Response and Defective Proteostasis

AD is a neurological disease characterized by the ubiquitous association of misfolded and aggregated proteins, whose role in the pathogenesis and progression of the disease is still unclear. However, it is reasonable to hypothesize that a significant dysfunction in protein homeostasis (proteostasis) occurs. Proteostasis is complex since it requires proteins to be in a specific localization, aggregation, concentration, and conformation. Multiple events occurring in AD have been suggested to act as proteostasis perturbators, including NFTs [86], neuroinflammation [87], altered calcium signaling [88], mitochondrial energy imbalance [89], and oxidative stress [90]. Most of these have been linked to endoplasmic reticulum (ER) stress [91]. The ER is an essential organelle in eukaryotes responsible for the synthesis and folding of all secretory and membrane proteins [92]. Under physiological conditions, when aberrant proteins are synthesized, the ER exports them to the cytosol, where they are directed to the ubiquitin–proteasome system for degradation [93]. In AD, the massive accumulation of aberrant misfolded proteins at the ER engages the unfolded protein response (UPR), a complex signaling system stress response that orchestrates protein folding and initiates apoptosis, or autophagy, in irreversibly damaged cells [94]. Growing evidence indicates that ER stress responses may also affect metabolic pathways that generate Aβ, suggesting its direct role in AD etiology. For instance, it has been demonstrated that UPR signaling events increase BACE1 levels, causing Aβ overproduction and promoting the transcription of the PSEN gene [95].

### 2.4. Complement Cascade and Neuroinflammation

Inflammation has been recognized as a key component of AD pathology [96], likely contributing even to the progression of the disease [97,98]. Several transcription factors involved in the inflammatory responses have been found involved in AD. For example, the CCAAT/enhancer-binding protein (c/EBP) family of transcription factors is elevated in brains from AD patients, compared to healthy controls [99], and it was found to promote microglial neuroinflammatory response [100]. Another example is the NF-kB pathway that controls cytokine production and cell survival, which is strongly associated with AD neuroinflammation [101].

Both the classical and alternative complement pathways are induced in vitro by fibrillar Aβ [102] and NFTs [103]. Senile plaques colocalize with microglia and many proteins of the complement cascade in animal models of the disease and human AD [62,104,105,106]. Moreover, human AD brains show signs of activation of the complement in the same areas presenting senile plaques and NFTs [107]. Complement factors have been shown to be elevated during AD progression, likely as a general reaction to abnormal protein deposition and other cerebral injuries that occur in the AD brain [108,109,110]. This is not surprising, since the complement cascade is a fundamental effector of the innate immune system that favors the rapid clearance of pathogens, apoptotic cells, and their debris, as well as the extent and termination of the inflammatory immune response [111]. Some components of the complement cascade play a key role in synapse pruning. This process is active and fundamental during the development of the nervous system. However, it is scarcely seen in the adult brain when its occurrence is thought to be detrimental, as in AD brains. Indeed, evidence of excessive complement-mediated synapse pruning has been reported in AD and animal models of aging [112,113,114]. Regardless, some human evidence shows inconsistency between blood and cerebrospinal fluid (CSF) concentration of complement proteins [110], highlighting the heterogenicity of the pathology, which complicates the path to use complement proteins as diagnostic biomarkers. However, components of the complement could be also potential novel therapeutic targets [111,115]. In preclinical models of neurodegenerative disorders, the inhibition of specific complement proteins had beneficial effects [116,117]. Unfortunately, the blood–brain barrier (BBB) is not accessible to current complement-targeted therapeutics, making drug design challenging [117]. Additionally, the molecular mechanisms underlying the inflammatory process observed in AD have not been fully clarified yet. This could explain the failure of the clinical trials conducted so far using conventional anti-inflammatory drugs [118,119,120,121,122].

Neuroinflammation is a complex defensive process crucial for the preservation of brain homeostasis that becomes detrimental under certain circumstances, which is not fully understood. It is now accepted that any cerebral insult triggers the activation of glial cells in a defensive, preservative process aimed at restoring the lost homeostasis. Both morphological and functional modifications of mainly, but not exclusively, microglia and astrocytes occur accompanied by a pro-inflammatory environment [19]. Microglia cells, being the immune sentinels of the central nervous system (CNS), are the first cells responding with a potent inflammatory response, consequently leading to the activation of other glial cell types, including astrocytes [123,124]. If the stimuli that activate glial cells are very intense, and/or long lasting, and/or not counterbalanced by an interruption signal, reactive gliosis could be established and the normal brain functioning could be altered, leading even to neuronal death [125]. However, the exact timing and mechanisms that turn neuroinflammation from a physiological to a pathological process are still under study [126,127]. Therefore, the clarification of the underlying molecular and cellular mechanisms could allow scientists to develop and test new, and hopefully efficacious, pharmacological treatments. For example, a recent study identified a negative regulator of the transcription factor c/EBPb, responsible for microglia-mediated neuroinflammation, which could represent a novel AD therapeutic target [100]. Of note, c/EBPb is expressed also by astrocytes. Thus, additional studies should address the possibilities of targeting it in different cell types involved in the neuroinflammatory process.

### 2.5. The Neuroenergetic Hypothesis

Glucose is the main brain energy fuel, which crosses the BBB through GLUT1, a membrane-bound glucose transporter. Both aging and AD are associated with a reduction of GLUT1 [128,129]. Additionally, transgenic mouse models show a correlation between the decreased density of GLUT1 and Aβ peptide accumulation [129,130]. In aged humans, an association between glucose hypometabolism and apoE genotype has been made [131]. The main signaling that mediates the uptake of glucose inside cells is the interaction of the pancreatic hormone insulin with its receptor. AD demented patients show a high level of plasma insulin, while low levels of both CSF insulin and brain insulin receptors. In accordance, insulin resistance has been correlated with dementia, and patients with type-2 diabetes have a much higher risk to develop AD [132]. Indeed, glucose acts as a memory enhancer since the neuronal activity is tightly coupled to glucose utilization [133]. Using 5xFAD mice as an AD model, Andersen et al. showed that neuronal GABA synthesis in the brain is directly affected by glucose hypometabolism in astrocytes [134]. Under normal conditions, astrocytes produce ATP and lactate that are released to feed neighboring neurons, in a process known as the astrocyte–neuron lactate shuttle, that energetically supports neurons given their high-energy requirements, such as action potential firing [135,136,137]. This shuttle is necessary for long-term potentiation [135]. Berchtold et al. reported that many genes involved in mitochondrial bioenergetics were upregulated in aged individuals with mild cognitive impairment (MCI), relative to age-matched controls, but downregulated in full-blown AD patients [138]. All this evidence contributed to the so-called neuroenergetic hypothesis, which posits that the chronic progressing starving of brain cells could produce energy-deficiency stress. This reduces neuronal firing and induces a shift from pathways associated with physiological APP metabolism to pathological ones, related to Aβ/tau production [139], ultimately leading to AD.

## 3. Astrocytes as Targets for AD Therapeutics

In the beginning, the interest in glial cells in AD arose mainly from the role played by the microglia cells in the immune response [140]. Afterward, it became clear that all types of glial cells were probably involved in both the etiology and progression of the disease, as actors in the context of the immune response and key regulating elements involved in the molecular and cellular processes altered in AD [141]. Indeed, cell-type-specific transcriptomic changes in human AD brains have been associated with distinct molecular pathways [142].

Glial cells are a heterogeneous cell population exerting a plethora of different actions necessary for the correct functioning of the brain [143]. Glial cells are usually classified into microglia and macroglia. The latter have a neural origin and include astrocytes, oligodendrocytes, and NG-2 glia, also known as synantocytes [144].

Microglia are the main immunocompetent cells of the nervous system with a non-neural origin. Being macrophages, they fulfill primarily defensive functions [145]. These cells regularly scan the surrounding environment with their processes and adapt their morphology and functions depending on what they sense. Upon activation, microglia exert chemotactic and phagocytic properties, moving where needed and clearing waste products, cellular debris, and pathogens [146]. In addition to these crucial defensive functions, microglia exert many other key actions related to synapse formation, pruning, and functioning [147,148,149]. Microglia cells show various activation states and expression profiles in both human AD brains and murine AD models [150]. The pathway analysis of single-nucleus transcriptomic experiments revealed that microglial genes mostly related to the immune response were differentially expressed between human AD brains and control subjects [142]. Additionally, the mutation in TREM2, a cell surface protein selectively and highly expressed by microglia in the brain, has been associated with a three-fold higher risk to develop AD [151].

Oligodendrocytes originate from precursor cells (OPCs) mainly localized in the ventricular zones of the brain, from which they migrate during development, through which they become mature oligodendrocytes. This process starts during the third trimester of gestation and continues throughout life [152]. Oligodendrocyte’s main function is the creation of the myelin sheath, crucial for effective neuronal transmission of action potentials [153]. Under the myelin sheath, in the internodal periaxonal space, oligodendrocytes establish direct connections with axons via cytoplasmic-rich myelinic channels, in which a bidirectional movement of macromolecules occurs between the two cells [152,154,155]. Impairments in myelin formation and functions have implications in several neurodevelopmental and neuropsychiatric disorders [156,157,158,159,160], and the maturation of OPCs into oligodendrocytes is accelerated by the loss of myelin due to injuries, aging, or diseases, including AD [157].

Astrocytes maintain CNS homeostasis at molecular, cellular, organ, and system levels of organization [161]. Several morphologically distinct subtypes of astrocytes have been identified that likely correspond to specific functions [162]. Indeed, they are present both in the white and grey matter. Astrocytes are key components of the BBB, thus regulating the communication between the CNS and the periphery [163]. They control the CNS microenvironment in several ways, including by buffering extracellular ions and the pH, regulating blood flow through the release of vasoactive molecules, and clearing reactive oxygen species (ROS) [164]. Astrocytes are components of the so-called gliocrine system, releasing around 200 molecules, mainly neurotrophic factors, and energy substrates, fundamental for the maintenance of CNS homeostatic functions [165]. Astrocytes exert primary roles in synaptic transmission and information processing by neural circuits. It has been demonstrated the ability of a single astrocyte to be in contact with several neurons and to modulate synaptic transmission by tuning neurotransmitter levels in the synaptic cleft [162,163].

Originally classified as OPCs, synantocytes are stellate cells, with large process arborizations that specifically express a new type of chondroitin sulfate proteoglycan [166]. They are found both in white and grey matter and interact with other glial cell types and neurons. Synantocytes extend processes along myelin sheaths to contact also the paranodes and nodes of Ranvier. Moreover, they were found to take part in the synaptic cradle, but their specific function at synapses has not been clarified yet [167,168].

Given the essential and pleiotropic functions driven by glial cells, the interest in the involvement of these cells in the pathophysiology of several neurological and neuropsychiatric disorders has grown exponentially in the last years [169]. Additionally, different glial cell types can communicate and influence each other’s phenotype and functions. However, the mechanisms and implications of those cross-talks are only beginning to be elucidated [124,170,171]. Below we focus on the evidence supporting a role for impaired astrocyte functioning in AD, and the potential therapeutic benefit that approaches aimed at restoring them could have.

The role of astrocytes in AD is difficult to decipher, mainly for two reasons: firstly, astrocytes exert a huge plethora of different functions in the CNS that are not easy to tease apart, and secondly, astrocytes respond to any perturbation of CNS homeostasis, caused by either injuries or diseases, with a variety of changes at structural, transcriptional, and functional levels. Additionally, the alterations are specific to the astrocyte localization and the CNS insult, and even to the different stages of the disease [125,172,173,174]. Regarding AD, the evidence available so far suggests the presence of both glial reactivity and atrophy since the initial stages of AD [97]. Additionally, astrocytes close to amyloid plaques show greater transcriptional changes than those far from plaques [175]. To complicate the picture, recent human studies showed that postmortem AD brains contain a reduced proportion of neuroprotective astrocytes, which are associated with glutamate recycling and synaptic signaling, compared to controls [142]. Moreover, the notion that astrocytes are asthenic in the final stages of AD is gaining ground. Regardless, both reactive and asthenic astrocytes operate in an erratic manner, thus contributing differently to the worsening of the disease through neuronal impairment and death [176]. Therefore, the difficulty to develop a pharmacological approach targeting astrocytes increases, since a drug directed to hypertrophic astrocytes in a specific AD stage could be detrimental in another stage at which astrocytes are atrophic, and vice versa. Moreover, modulating astrocytes could affect the functioning of other glial cell types, besides neurons [177,178], altering the normal communication among brain cells. Another important challenge to overcome when designing a therapy directed to the brain is the necessity for it to cross the BBB. It has been reported that only 5% of about 7000 drugs screened in the Comprehensive Medical Chemistry database are actually able to enter the CNS passing the BBB [179,180].

There is a growing number of reports looking at the role of astrocytes in AD, and several approaches targeting astrocytes have been proposed (Figure 1). The following sections review both in vitro and in vivo evidence that has been published in the last five years targeting astrocytes pharmacologically in models of AD (Table 1).

### 3.1. Targeting Astrocyte Senescence

Aging is considered one of the main risk factors for the development of neurodegenerative diseases, including AD [220]. Studies on cellular aging are attracting much attention as a fervent area of research [221,222], and recent evidence demonstrates that astrocytes senescence has a critical role in the pathogenesis of AD. As time goes by, astrocytes show peculiar cellular and molecular changes assuming the so-called senescence-associated secretory phenotype (SASP) [223]. This is accompanied by upregulation and release of proinflammatory cytokines, including interleukin(IL)-1β and IL-6, chemokines, and proteinases [175,184,224]. Overexpression of intermediate filament proteins glial fibrillary acidic protein (GFAP) and vimentin occurs, whereas neurotrophic growth factors result downregulated. The chromatin undergoes several modifications, and there is upregulation of p53, p21^WAF1^, and p16^INK4A^, leading to a permanent cell cycle arrest [225,226]. These features may not be specific senescence markers for astrocytes since they are postmitotic cells that do not usually divide in healthy tissues [126]. Regardless, one of the most common features of aging is the accumulation of senescent cells. Bussian et al. demonstrated that the presence of senescent astrocytes and microglia in a mouse model of aggressive tauopathy (the *PS19* mice) promotes the formation of hyperphosphorylated tau aggregates. Removing p16^INK4A^-expressing senescent cells through a genetic approach prevented astrogliosis, hyperphosphorylation of tau, degeneration of cortical and hippocampal neurons, and it preserved transgenic mouse cognitive functions [182]. Comparable effects have been obtained by testing a senolytic agent, the orally active anticancer drug ABT263 (navitoclax), that acts as inhibiting Bcl-2. By this mechanism, this compound is able to induce apoptosis specifically in senescent cells [227]. The clearance of accumulated senescent astrocytes also rescued in vivo the radiation-induced impaired astrocytic neurovascular coupling and mice cognitive performance [181]. Another report showed that the overexpression of an inhibitory isoform of p53, the Δ133p53, which is downregulated in AD, repressed the SASP after its induction in astrocytes by exposure to radiation. Δ133p53 overexpression promoted also DNA repair and inhibited irradiated astrocyte-mediated neuroinflammation and neurotoxicity [183]. The antiprotozoal drug pentamidine upregulates p53 and increases the ratio BAX/Bcl2, ultimately promoting apoptosis in cultured astroglioma cells [228], and it exerts anti-inflammatory effects in mice receiving human Aβ42 into the hippocampus [229]. Finally, from the field of phytotherapy, an in vitro study showed that Ginsenoside F1 suppresses the SASP in astrocytes by downregulating the p38MAPK-dependent NF-κB activity [184], a pathway upregulated in AD.

### 3.2. Targeting Astrocyte Glutamate Transporters

Glutamate represents the major excitatory neurotransmitter of the CNS, whose neurotransmission is finely regulated by both neurons and glial cells [230]. Astrocytes, in particular, are responsible for glutamate reuptake from the synaptic cleft through excitatory amino acid transporters (EAATs). There are five subtypes of EAATs (EAAT1–EAAT5), but EAAT2 (glutamate transporter-1/GLT1) is responsible for more than 90% of glutamate reuptake [231]. Once inside the astrocyte, glutamate is converted mainly into glutamine by the glutamine synthetase (GS) and then shuttled back to the presynaptic neuron, which uses it to synthesize glutamate again. A portion of glutamate is converted to gamma-aminobutyric acid (GABA), which is usually catabolized. The glutamate–glutamine shuttle is crucial for glutamate homeostasis, and thereby for learning and memory. If the shuttle is dysfunctional, an abnormal glutamate stimulation could occur, which is neurotoxic [232]. Glutamate excitotoxicity has been observed in AD and correlated with cognitive decline [232,233]. In parallel, both accumulation of GABA, whose concentration is low in astrocytes under physiological circumstances [234], and its release from reactive astrocytes have been observed in transgenic animal models of AD (5xFAD and APP/PS1 mice), resulting in memory deficits [235,236]. However, astrocytic GABA content seems to follow a bell-shaped curve along aging and not relate to Aβ [237]. Human postmortem AD brains showed altered expression of several GABA transporters in cortical and hippocampal regions [238]. Therefore, counteracting dysfunctions in the content of neurotransmitters and the expression of their transporters could likely be beneficial in AD. Research targeting the modulation of astrocytic GABA is still not fully explored, and further studies are warranted. Instead, the enhancement of glutamate transporter function and expression has been tested using various activators in several neurological diseases [239]; however, few studies were carried out in AD models. β-lactam antibiotics are drugs that upregulate GLT1 gene transcription, in addition to having antibacterial effects [240,241]. Among them, ceftriaxone was found to ameliorate AD pathology by improving spatial learning and memory in APP/PS1 mice, upregulating the expression of both GS and the system N glutamine transporter 1 (SN1) [185]. Another drug already approved for human use is riluzole, which has been shown to improve memory performance in aged rats and in 5xFAD mice [186,242]. Riluzole is a neuroprotective agent able to increase Na^+^-dependent glutamate uptake in synaptosomes in a dose-dependent manner [243]. Riluzole chronic oral administration prevents age-related gene expression changes in rats’ hippocampi [244] and reduces the levels of Aβ42 and Aβ40 oligomers and neuritic plaques in 5xFAD mice [186]. Despite being so promising, these results have not been translated into the clinic yet.

### 3.3. Targeting the Astrocytic Metabolic System

As we mentioned before, failure of astrocytes in supporting neuronal energy needs could facilitate the progression from physiological to pathological brain aging. For instance, the metabolic products of fatty acid oxidation decrease during AD [245], making lipid metabolism a potential target for AD treatment. Recently, an in vitro study found that activation of the peroxisome proliferator-activated receptor proliferator-activated receptor (PPAR) beta/delta (PPARβ/δ) increases fatty acid oxidation [187]. Indeed, a rate-limiting enzyme of the fatty acid oxidation is the carnitine palmitoyltransferase 1A (CPT1A), which catalyzes the transfer of fatty acids into the mitochondria, where the β-oxidation occurs. Konttinen et al. tested the effects of GW0742, a synthetic PPARβ/δ agonist, in human astrocytes obtained from pluripotent stem cells (iPSCs) of AD patients carrying an amyloidogenic mutation of PSEN1 (PSEN1ΔE9). GW0742 enhanced the expression of CPT1a, increasing astrocyte fatty acid oxidation [187]. In primary astrocytes obtained by 5xFAD mice, which show an altered metabolic profile, administration of the vitamin B5 precursor pantethine reversed several metabolic alterations induced by Aβ challenge, including (i) altered activity of the glucose-6-phosphate dehydrogenase, the α-ketoglutarate dehydrogenase complex, and the succinate dehydrogenase; (ii) decreased ATP production; and (iii) altered expression of the hypoxia-inducible factor-1 alpha, known to protect against Aβ toxicity. Pantethine treatment showed some anti-inflammatory actions by downregulating IL-1β expression [188]. Similarly, treatment of the astroglioma cell line C6 with hydroxytyrosol, the major polyphenol contained in olives, ameliorated the metabolism of glucose, previously altered by Aβ(25–35) challenge, through activation of Akt [189]. Evidence demonstrates the ability of glucagon-like peptide-1 (GLP-1) to improve cognitive deficits in AD [246]. Zheng et al. just published that this effect is related to GLP-1 ability to restore in vitro the Aβ-induced glycolysis impairment in astrocytes, by activating the PI3K/Akt pathway [190]. A recent study by Wang et al. demonstrated that metformin, a hypoglycemic drug of clinical use, exerts anti-inflammatory and antioxidant effects in rat primary astrocytes treated with high glucose concentration [191], strengthening the link between altered metabolism and induction of inflammatory process.

### 3.4. Upregulation of Astrocytic Neurotrophins and Growth Factors

Neurotrophic factors imbalance and dysregulation are associated with neurodegenerative diseases, including AD [247]. The brain-derived neurotrophic factor (BDNF) is involved in cognition and memory formation, given its role in modulating synaptic plasticity. Astrocytes can release neurotrophic growth factors, including BDNF, exerting protective effects on neurons [248]. Thus, the increase in astrocyte neurotrophic factor expression and release could be a therapeutic approach for AD [249]. Sawamoto et al. found that the citrus flavonoid 3,5,6,7,8,30,40-heptamethoxyflavone (HMF) exerts neuroprotective effects by increasing the expression of BDNF in astrocytes within the hippocampus of mice and in the C6 glioma cell line. The BDNF increase was induced by the activation of cAMP/ERK/CREB signaling and inhibition of phosphodiesterase 4B and 4D [192]. Another molecule found to be able to upregulate BDNF expression in cultured astrocytes is quetiapine, a widely used atypical antipsychotic drug [193]. Recently, a paper in which transgene delivery in astrocytes was used to obtain the upregulation of BDNF in 5xFAD mice was published [194]. Specifically, 5xFAD mice were crossed with transgenic pGFAP-BDNF mice, expressing BDNF under the GFAP promoter. The resulting transgenic mice showed restored levels of BDNF, compared to 5xFAD mice, which have reduced levels of this neurotrophin compared to their wild-type counterparts. BDNF restoration also resulted in improvements in cognitive tasks and ameliorated synaptic plasticity [194].

Some studies have also explored the potential beneficial effects of neural stem cell transplantation in models of AD. An Indian group studied the lineage negative stem cells (Lin-ve) derived from human umbilical cord blood (hUCB) in an animal model of Aβ42-induced injury. They found that intrahippocampal transplant of these cells at specific dosage and timing shows potential to reverse hippocampal Aβ42-induced mouse cognitive impairment, measured by Morris water maze and passive avoidance, through a neuroprotective mechanism likely mediated by BDNF upregulation [195,250]. Blockade of the BDNF-TrkB pathway by systemic administration of a TrkB inhibitor nullified the benefit of Lin-ve cell transplant. Aβ42-challenged mice showed decreased BDNF and GFAP protein and gene expression, which were both reversed by Lin-ve cell transplant. Some less clear effects were detected also in the expression levels of both the glial-derived neurotrophic factor (GDNF) and the ciliary neurotrophic factor (CNTF), which deserve further studies [195].

AD pathogenesis is also affected by altered production of growth factors [251,252], including the fibroblast growth factor (FGF) 2 [247]. In particular, FGF2 is increased in reactive astrocytes around senile plaques [253]. Last year, Chen et al. demonstrated that FGF2 has protective effects against Aβ42-induced cytotoxicity in primary cultured cortical astrocytes. In their experiments, primary astrocytes challenged with Aβ42 were treated with either high or low molecular weight forms of FGF2. The low molecular isoform of FGF2 promoted astrocyte proliferation, enhancing the expression of c-Myc, Cyclin D1, Cyclin E [196].

### 3.5. Targeting Astrocytes-Driven Amyloid Aggregation and Clearance

Accumulation of Aβ could be the result of its increased synthesis or reduced clearance or a combination of both. Looking for AD treatment, an important area of investigation targets Aβ clearance, which depends, at least in part, on astrocytes. Indeed, astrocytes can take up Aβ and digest it in their lysosomes. However, the astrocytic degrading machine could get engulfed, leading to detrimental consequences [254]. Lysosome functions and gene expression for proteins involved in both autophagy and proteolysis were found altered in aging and AD [255,256]. Two of the apolipoproteins associated with high risk for developing sporadic AD are secreted by astrocytes and are involved in Aβ aggregation and clearance, the apoE4 and the apoJ (also known as clusterin) [69,257]. The apoE4, once secreted by astrocytes, binds to high-density lipoprotein (HDL)-like particles, and the level of its lipidation influences Aβ aggregation and clearance [258]. Chernick et al. demonstrated the ability of an HDL mimetic peptide, the 4F, to increase apoE4 lipidation in primary human and murine astrocytes. That counteracts the Aβ-induced accumulation of intracellular apoE4, mitigating Aβ detrimental effects on proper cellular trafficking and functionality of apoE [197]. Clusterin (Clu) is a ubiquitous protein whose functions are still not clear, but studies have shown its involvement in Aβ aggregation, toxicity, and clearance. Conflicting results have been published reporting both neuroprotective and detrimental properties of Clu [259,260]. Novel in vitro findings demonstrated a role for astrocytic Clu in promoting synapse formation and glutamatergic synaptic activity [199]. Wojtas et al. overexpressed Clu (>about 30%) selectively in GFAP-positive astrocytes of APP/PS1mice and noticed a reduction in Aβ accumulation and formation of fibrillary deposits in both cortex and hippocampus compared to control animals. In the same brain areas, the authors found that Clu overexpression was associated with a reduction of the number of cortical and hippocampal dystrophic neurites [198]. In accordance, the reduction (<about 50%) in Clu expression in GFAP-positive astrocytes of APP/PS1 mice leads to a worsening of the AD-like outcomes [198]. Novel in vivo findings demonstrated that Clu overexpression in astrocytes enhances excitatory neurotransmission and rescues the synaptic deficit in Clu knockout mice. Clu overexpression in GFAP-positive astrocytes of 5xFAD mice reduced plaque numbers and plaque size and rescued presynaptic dysfunction [199].

Another molecule that seems to promote Aβ clearance is the epigallocatechin gallate (EGCG), a member of the catechin family. In cultured astrocytes, ECGC elevates neprilysin (NEP) expression, one of the most important Aβ-degrading enzymes in the brain, involving also the activation of ERK and phosphoinositide 3-kinase [200].

Moreover, oral administration of fish oil, containing n-3 polyunsaturated fatty acids (PUFAs), was found effective in clearing Aβ from the brain of fat-1 transgenic mice [201], but not of aquaporin (AQP) 4 knockout mice, suggesting the involvement of AQP4 protein, expressed selectively in astrocytes, in Aβ clearance. Additionally, PUFAs administration protected from AQP4 polarization occurring after Aβ injection [201], a sign of astrocytic dysfunction [261].

### 3.6. Targeting Astrocytic Reactivity, Complement Cascade, Neuroinflammation, and Oxidative Stress

Neuroinflammation plays a pivotal role in the development and progression of AD. Indeed, Aβ plaques are surrounded by activated glial cells, and Aβ itself leads to the activation of astrocytes and microglia, together with the release of proinflammatory factors [97,262,263,264]. Brains of different transgenic mouse models of AD show activated astrocytes, even before the appearance of plaques and NFTs [265,266]. When astrogliosis occurs, reactive astrocytes produce inflammatory markers, such as tumor necrosis factor (TNF)-α, IL-1β, and IL-6, and calcineurin, a protein phosphatase that mediates inflammatory responses. This is associated with a wide number of cellular events, including the aforementioned activation of the complement cascade, the release of nitric oxide, and ROS. This phenomenon is normally engaged with the intent of defending the brain by removing injurious stimuli (e.g., Aβ fibrils phagocytosis). However, if prolonged beyond physiological limits, it would have detrimental effects. Therefore, targeting astrocyte reactivity and, consequently, the related activation of the complement cascade, the oxidative stress and the inflammatory response could represent an effective therapeutic strategy in AD. A compound that has shown such properties is cannabidiol, the main nonpsychoactive component of Cannabis Sativa [267]. Studies demonstrated cannabidiol effects in reducing both GFAP and S100B mRNA and protein expression, as well as neuroinflammatory parameters in different models of AD [268,269,270].

The complement component C3 is increased in human AD brains, and it is expressed by reactive astrocytes. Its increased expression is required for neurodegeneration to occur [271]; thus, its targeting could be beneficial. Indeed, Shi et al. compared aged plaque-rich transgenic APP/PS1 mice knockout (KO) for the C3 to transgenic APP/PS1 mice to evaluate Aβ plaque pathology, glial responses to plaques, and neuronal dysfunction in the brains. They found that C3 KO mice had less activation of glial cells at the center of Aβ plaques compared to control mice, suggesting that the downregulation of C3 controls astrocyte activation and neuroinflammation in AD [202].

Mc Manus et al. tested the effect of infection by Bordetella Pertussis in APP/PS1 mice and the potential benefit of fingolimod (FTY720) administration, an FDA-approved immunomodulatory drug for treating multiple sclerosis. Fingolimod reduced signs of infection-induced BBB increased permeability, GFAP immunoreactivity, and Aβ deposits, compared to control mice. Results of additional in vitro experiments in primary astrocytes suggested that the decreased Aβ accumulation was driven by the fingolimod-induced increase in the phagocytic capacity of astrocytes [203].

Since Aβ activates the astrocytic inflammasome promoting the release of IL-1β, Couturier et al. demonstrated that the downregulation of this Aβ-induced inflammatory process increases Aβ phagocytosis in astrocytes in vitro. That is due to the release of the chemokine CCL3, ultimately improving in vivo the memory deficits of 5xFAD mice [204]. Therefore, that phlogistic event represents a druggable therapeutic target, which still needs to be thoroughly investigated. Several molecules have been tested during the last years for their ability to dampen astrocyte reactivity in AD [269,270,272,273], but none have been translated to the clinic yet. Patients with MCI and vascular dementia show increased levels of Lipocalin 2 (Lcn2) in the CSF. In AD cases (stages V and VI), Lcn2 immunoreactivity increased in reactive astrocytes localized around plaques and in reactive microglia [274]. Astrocytes are the major producers of Lcn2 in the brain [275]. This protein is involved in several processes including inflammation, iron metabolism, cell death, and cell survival, modulating the cellular response to Aβ [275]. Staurenghi et al. demonstrated that increased levels of oxysterols observed in mild or severe AD brains, accompanied by increased levels of Lcn2, determine a clear morphological change in mouse primary astrocytes [276]. A recent study found that the iron chelators deferoxamine and deferiprone reduce Aβ-induced iron accumulation in astrocytes and inhibit Aβ-induced Lcn2, suggesting these molecules as promising therapeutic strategies against AD [205]. A novel synthesized compound, Glu-DAPPD, containing a glucose group linked to an anti-neuroinflammatory agent, the N,N′-diacetyl-p-phenylenediamine, showed in vivo to reduce Aβ aggregates and immunostaining for astrocytes and microglia, and to improve cognitive function in transgenic APP/PS1 mice being administered chronically for two months [206].

Recent studies identified the Janus kinase 2-signal transducer and activator of transcription 3 (JAK2-STAT3) pathway as a key pathway for the induction and maintenance of astrocyte reactivity. Using adenoviral delivery techniques, authors either downregulated or upregulated the JAK2-STAT3 pathway specifically in hippocampal astrocytes. They found that the JAK2-STAT3 pathway is necessary and sufficient to trigger astrocyte reactivity in the hippocampus of transgenic APP mice, controlling also for gene expression of a variety of genes, of which many involve the inflammatory process. The downregulation of this pathway reduced also Aβ deposits and improved mice spatial learning but not memory retrieval. On the other hand, the upregulation of the JAK2-STAT3 pathway resulted in opposite and deleterious results [207].

Astrocytes are involved in both the production and clearance of ROS, concurring to the oxidative stress found in AD, whose reduction has been tested as a potential therapeutic target. Interestingly, mobile phone radiofrequency electromagnetic fields (EMF) have been shown to reduce both Aβ and H_2_O_2_-induced ROS production in human and rat primary astrocytes, as well as the co-localization between the cytosolic (p47-phox) and membrane (gp91-phox) subunits of NADPH oxidase, indicating the suppression of its activity [212]. Other antioxidant anthocyanin compounds have recently been investigated [277]. Among them, pelargonidin, which acts as an estrogen receptor agonist, has been tested in rats that received an intrahippocampal injection of Aβ(25–35). Pelargonidin treatment resulted in improved Morris water maze test performance. Higher hippocampal catalase and acetylcholinesterase activities have been detected, accompanied by lower GFAP protein expression, but no change in inducible nitric oxide synthase (iNOS), compared to control animals [213].

Recently, the compound monascin has been found to activate the expression of several antioxidative genes such as SOD-1, SOD-2, SOD-3, and HSP16.2 and reduce Aβ-toxicity in C. elegans strain [214], suggesting its antioxidant potential. In addition, resveratrol [278], tocotrienol [279], epicatechins [280], H-1,2-dithiole-3-thione [281], curcumin, and epigallocatechin-3-gallate [282] have shown in vitro and in vivo anti/oxidant properties in several models of Aβ-mediated toxicity and AD.

As fundamental regulators of brain homeostasis, astrocytes also regulate the intracellular Ca^2+^ concentration through an intermediate conductance calcium-activated potassium channel, KCa3.1. This channel is actively involved in the phenotypic change of astrocytes during astrogliosis observed in AD. By using KCa3.1 knockout mice, memory deficits, neuronal loss, glial activation, tau phosphorylation, and insulin signaling deficits were ameliorated compared with control animals, making this channel an interesting pharmacological target in AD [215]. During the neuroinflammatory process, ATP and ADP are released around plaques, leading to the activation of the metabotropic P2Y1 purinoreceptors (P2Y1Rs) expressed by astrocytes, which increases the rate of spontaneous calcium events [283]. Chronic intracerebroventricular infusion of P2Y1R inhibitors resulted in structural and functional restoration of astrocytes and the preservation of memory deficits [216].

Since AD patients show increased levels of the Gs-coupled adenosine receptor A_2A_ in astrocytes, Orr et al. studied in vivo the ablation of astrocytic A_2A_ receptors demonstrating that it enhances long-term memory [284]. The adenosine tone on the astrocytic A_2A_ receptors has also been modulated through a new BBB-permeable equilibrative nucleoside transporter (ENT) inhibitor, J4, tested in APP/PS1 mice. In particular, J4 inhibited the recycling of adenosine from the extracellular space performed by ENTs, resulting in the prevention of the decline in spatial memory, a common feature in AD patients [217]. Additionally, istradefylline, a selective antagonist of A_2A_ receptors, enhanced the performance in behavioral tests in transgenic APP mice [218].

### 3.7. Modulation of Astrocytes According to Their Morphofunctional State: The Case of Palmitoylethanolamide

In AD, as in other neurodegenerative disorders, astrocytes undergo morphological, biochemical, metabolic, and transcriptional changes, as well as physiological remodeling. All these rearrangements could lead to either a gain or loss of one or more functions [126]. Thus, pathological changes of astrocytes should not just refer to hypertrophy. Indeed, also morphological atrophy could contribute to AD early synaptic failures and cognitive deficits [126,285]. For these reasons, molecules able to modulate astrocyte morphology and functions according to their reactive or atrophic status could be potentially valuable therapeutics. To the best of our knowledge, the only molecule that has so far shown some indications to exert such effects is palmitoylethanolamide (PEA).

PEA is a naturally occurring amide of ethanolamide and palmitic acid, firstly isolated from soy lecithin, egg yolk, and peanut meal. It acts as a lipid messenger that mimics several endocannabinoid-driven actions, even though it does not bind to cannabinoid receptors [286]. We and other groups have shown that PEA exerts anti-inflammatory and neuroprotective properties in several preclinical models of Aβ-induced toxicity and AD [287]. PEA in vitro attenuates Aβ-induced astrocyte expression of GFAP and S100B and the release of pro-inflammatory molecules [273,288]. In a surgical model of Aβ-neurotoxicity PEA treatment reduced astrocyte hypertrophy and markers of inflammation, including iNOS, cyclooxygenase (COX)-2, IL-1β, and TNF-α [289]. PEA also demonstrated the ability to protect Aβ-induced neuronal reduced viability and loss in vitro, ex vivo, and in vivo [208,209,286,289,290]. These results have been confirmed also in primary astrocytes derived from the prefrontal cortex of 3xTg-AD mice, in which PEA promoted neuronal viability [210]. All these reports concurred to demonstrate that PEA exerted these effects through the PPARα by the use of selective antagonists, corroborated by experiments in models where the receptor was genetically ablated [291,292,293]. However, studies showed that PEA effects could involve also the orphan G-protein coupled receptor 55 [294], and the transient receptor potential vanilloid type 1 channel [295]. Moreover, PEA is able to exert an indirect activation of cannabinoid receptors, via the so-called entourage effect [296], working as a false substrate for fatty acid amide hydrolase, an enzyme involved in the metabolism of the endocannabinoid anandamide (AEA) [297]. Indeed, due to the reduction of its catabolism, AEA levels rise. Thus, in turn, AEA could bind to cannabinoid receptors. One additional peculiar feature of PEA is its ability to act as an autacoid local injury antagonist, thus dampening mast cells that are now considered critical effectors during AD progression [298]. In this way, PEA contributes to protecting neurons from excitotoxicity [297]. Interestingly, the modulation of the cross talk between mast cells and glial cells is emerging as a valuable approach to treat several neuroinflammatory brain pathologies, including AD [299]. Some articles present an extensive review of PEA biological functions in the CNS [296,297,300].

Different formulations of PEA have been synthesized to improve its bioavailability and efficacy, including the ultramicronized (um-PEA) and PEA-oxazoline forms as well as the combination of PEA with luteolin (Lut), an antioxidant compound, ultramicronized together (co-ultra PEA/Lut). Pretreatment with um-PEA of rat hippocampal slices challenged acutely with Aβ42 significantly reduced iNOS and GFAP expression [301]. It also restored cell viability of glioma and neuroblastoma impaired by lipopolisaccaride and interferon-gamma treatment, reducing protein expression of both iNOS and COX-2 [211]. Um-PEA demonstrated oral bioavailability and its chronic administration reduced neuroinflammatory markers and showed neuroprotective effects in 3xTg-AD mice [210,219,302,303]. When comparing hippocampi of 6-month-old with 12-month-old 3xTg-AD mice, the younger animals did not show astrocyte hypertrophy (measured as an increase in GFAP immunoreactivity) but exhibited an ongoing intense neuroinflammatory process with high levels of iNOS, TNF-α, chemokines, and interleukins, whereas older mice showed significant astrocyte atrophy without elevation in neuroinflammatory markers. Chronic subcutaneous pretreatment with um-PEA for 3 months prevented the establishment of the phlogistic process in hippocampi of 6-month-old 3xTg-AD mice, compared to vehicle-treated ones. Um-PEA also prevented the altered performance in cognitive tasks and reduced Aβ formation and phosphorylation of tau protein in the hippocampus [219]. Astrocyte hypertrophy was detected in the cortices of vehicle-treated 6-month-old mice, and um-PEA chronic treatment decreased both GFAP mRNA and protein expression [210]. Interestingly, 3xTg-AD mice that received um-PEA subcutaneous administration for 3 months, before being tested at 12 months of age, showed restored astrocyte GFAP immunoreactivity to the level of non-Tg controls, also improving their outcome in behavioral assessment of short-term memory [219]. Collectively these reports show that um-PEA acted preventing either astrocyte hypertrophy either atrophy. This indicates that PEA behaved as a modulator of astrocyte morphology and cell reactivity state. This is in accordance with the current view seeing astrocyte reactivity as an evolving and reversible process caused by extrinsic triggers [126,304].

Another formulation that combines the aforementioned PEA effects with the antioxidant actions of Lut has been tested in preclinical AD models. Co-ultra PEA/Lut showed anti-inflammatory and antiapoptotic effects in Aβ42-challenged rat hippocampal slices and neuroblastoma cells [301]. In vivo, co-ultra PEA/Lut administration for two weeks in rats that received a single intrahippocampal infusion of Aβ42 prevented the Aβ-induced astrocyte hypertrophy, as well as the upregulation in gene expression of pro-inflammatory cytokines and enzymes found in rats treated with vehicle. Moreover, co-ultra PEA/Lut prevented the Aβ-mediated decrease in gene expression of both glial-derived and brain-derived neurotrophins [35]. Despite having these promising features, no studies have yet elucidated the synergic mechanisms of actions of the association of PEA with Lut. Regardless, since co-ultra PEA/Lut administration started the same day of the surgical infusion, to model the very first phase of Aβ42 accumulation as in the prodromal stage of AD, the above-reported study mimicked a potential therapeutic intervention at the earliest stage of the disease. The results support the thesis that targeting astrocytes at the beginning of the pathology could have a positive impact. Other very recent studies endorse this view. Reports from Dr. Escartin’s group modulated the activation of astrocytes in 9-month-old 3xTg-AD mice. The downregulation of the JAK2-STAT3 pathway fully restored mice early synaptic and long-term potentiation alterations [207], improved short-term memory, and reduced anxiety behavior [176], thus supporting the hypothesis that targeting astrocytes at the very early stages of AD could be beneficial.

The potential translational value of ultramicronized or co-micronized PEA as a preventive therapeutic strategy in AD is corroborated by its safety and tolerability, as it is already in the human and veterinary market as food for special medical purposes and complementary feed, respectively. Some single or few-cases human studies have been carried out showing favorable results in improving MCI and frontotemporal dementia [305,306], in recovering from stroke [307], and in managing neuropathic pain associated with neuroinflammation [308].

## 4. Conclusions

Despite the spasmodic basic and medical research and the existence of approved therapies, there is a huge unmet clinical need for effective therapies for AD, especially treatments that are intended to address the biological basis of the pathology to favorably modifying its long-term course. Currently approved drugs do not target the underlying pathology of AD since they only provide modest beneficial effects to a small subset of patients. Moreover, no treatments are available to counteract AD at its earliest stage, which could represent the best timepoint to start therapy. Indeed, Aβ deposition into amyloid plaques, followed by markers of neurodegeneration, tau pathology, and reduction of brain volume, initiates decades before the onset of observable clinical signs. Dysfunctions of astrocytes have been linked to the molecular alterations observed in AD, thus representing a promising target for disease management. However, morphofunctional changes occurring in astrocytes vary depending on the stage of the pathology. Therefore, molecules capable of correcting dysfunctions of astrocytes could represent a promising pharmacological strategy. Reviewing the literature findings, the only compound so far that seems to exert this effect is PEA. Our previous study indeed showed the ability of PEA to normalize the astrocyte alterations observed in an experimental model of AD, the 3xTg-AD mice, endowed with face, construct, and predictive validities, bringing them back to a homeostatic condition. That and other possibilities of new therapeutic approaches represent an important springboard for the development of therapies for a still incurable disease, such as AD.

## Figures and Tables

**Figure 1 biomolecules-11-00600-f001:**
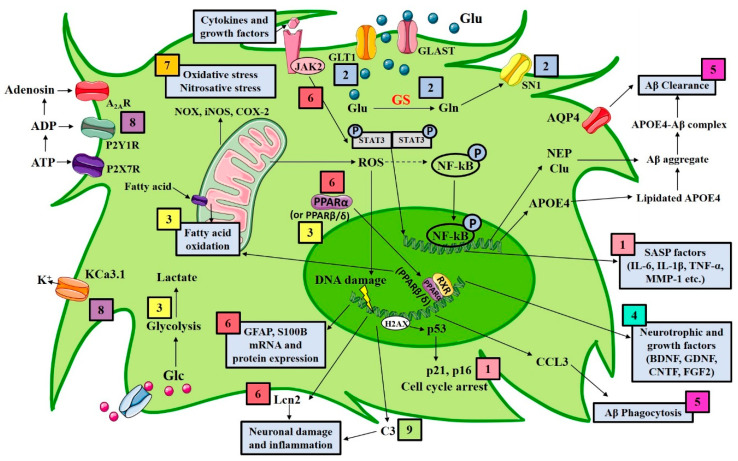
Schematic representation summarizing different molecular mechanisms of astrocytes to be manipulated in AD. The figure shows the main astrocytic pharmacological targets for the treatment of AD: (1) astrocyte senescence; (2) glutamate transporters; (3) astrocytic metabolic system; (4) upregulation of neurotrophins and growth factors; (5) astrocytic amyloid clearance and phagocytosis; (6) astrocytic reactivity; (7) astrocytic oxidative stress; (8) astrocytic channels and receptors; (9) astrocytic complement cascade. A2AR, adenosine 2A receptor; Aβ, amyloid β; ADP, adenosine diphosphate; APOE4, apolipoprotein E4; ATP, adenosine triphosphate; AQP4, aquaporin 4; BDNF, brain-derived neurotrophic factor; CCL3, C-C motif chemokine ligand 3; Clu, clusterin; CNTF, ciliary neurotrophic factor; COX-2, cyclooxygenase-2; FGF2, fibroblast growth factor 2; GDNF, glial-derived neurotrophic factor; GFAP, glial fibrillary acidic protein; GLAST, glutamate aspartate transporter; Glc, glucose; Gln, glutamine; GLT-1, glutamate transporter-1; Glu, glutamate; GS, glutamine synthetase; H2AX, histone family member X; IL-1β, interleukin 1β; IL-6, interleukin 6; iNOS, inducible nitric oxide synthase; JAK2, janus kinase 2; KCa3.1, calcium-activated potassium channel 3.1; Lcn2, Lipocalin 2; MMP-1, matrix metalloproteinase-1; NEP, neprilysin; NF-kB, nuclear factor-kB; NOX, NADPH oxidase; PPARα, peroxisome proliferator-activated receptor α; PPARβ/δ, peroxisome proliferator-activated receptor β/δ; P2X7, purinergic receptor; P2Y1R, P2Y1 purinergic receptor; ROS, reactive oxygen species; RXR, retinoid X receptor; SASP, senescence-associated secretory phenotype; S100B, S100 calcium-binding protein B; SN1, N glutamine transporter 1; STAT3, signal transducer and activator of transcription 3; TNF-α, tumor necrosis factor α.

**Table 1 biomolecules-11-00600-t001:** In vitro and in vivo approaches targeting astrocytes in Alzheimer’s disease.

Astrocytic Target	Experimental Strategy	Results	References
*Senescence*	Removal of senescent cells in vivo by radiation treatment or by genetic ablation	Reduction in astrogliosis, tau hyperphosphorylation, neuronal degeneration; preservation of cognition	[181,182]
	*In vivo* administration of the senolytic drug ABT263 (navitoclax)	Prevention from the upregulation of senescence-associated genes attenuated tau phosphorylation; cognitive improvements	[181,182]
	Overexpression of Δ133p53 in radiation-induced senescent astrocytes	Repression of the irradiation-induced SASP	[183]
	Ginsenoside F1 in vitro treatment	SASP suppression by downregulation of p38MAPK-dependent NF-κB pathway	[184]
*Glutamate transporters*	Ceftriaxone administration in APP/PS1 mice	Raise in GLT1, GS and SN1 protein expression and cognitive performance improvements	[185]
	Chronic oral administration of riluzole in 5xFAD mice	Prevention of senescent associated gene expression changes; reduction of Aβ oligomers and plaques	[186]
*Metabolism*	PPARβ/δ agonist treatment of human AD astrocytes (PSEN1ΔE9)	Enhancement of AD-reduced fatty acid oxidation	[187]
	Pantethine in vitro treatment of astrocytes obtained from 5xFAD mice	Reversal of the altered activity of several metabolic enzymes and of the induced IL-1β expression	[188]
	Hydroxytyrosol treatment of glioma cell cultures challenged with Aβ (25–35)	Proper glucose metabolism restoration by Akt activation	[189]
	GLP-1 in vitro treatment of Aβ-exposed astrocytes	Reversal of the Aβ-altered glycolysis by activation of the PI3K/Akt pathway	[190]
	Metformin in vitro treatment of astrocytes exposed to high glucose concentration	Inhibition of both the ER stress and inflammation induced by high glucose	[191]
*Neurotrophins and growth factors*	HMF treatment of primary astrocytes and C6 glioma cell line	Raise in BDNF expression induced by both the activation of cAMP/ERK/CREB signaling and the inhibition of PDE4B and PDE4D	[192]
	Primary neurons exposed to Aβ (25–35) incubated with quetiapine-treated astrocyte conditioned medium	High BDNF release by astrocytes treated with quetiapine promoted viability of primary neurons	[193]
	Overexpression of BDNF specifically in GFAP-positive astrocytes by genetic crossing in 5xFAD mice	The raise in BDNF levels that are reduced in 5xFAD mice improved synaptic plasticity and cognition	[194]
	*In situ* stem cell transplant in intrahippocampal Aβ42 infused mice	Reversal of the Aβ42-induced cognitive impairment by BDNF-TrKB pathway activation	[195]
	FGF2 treatment of primary astrocytes challenged with Aβ42	Promotion of astrocyte proliferation through enhanced expression of c-Myc, Cyclin D1, Cyclin E	[196]
*Aβ clearance*	HDL mimetic peptide in vitro treatment of primary human and murine astrocytes	Raise in apoE4 lipidation lowers its detrimental cellular accumulation	[197]
	In vivo overexpression or downregulation of Clu specifically in GFAP-positive astrocytes in APP/PS1 mice	Clu overexpression is associated with a reduction in Aβ burden. The opposite phenomenon was found with Clu downregulation	[198]
	In vivo overexpression of Clu specifically in GFAP-positive astrocytes in 5xFAD mice	Reduction in plaques number and sizes. Improvement in synaptic function	[199]
	EGCG treatment of Aβ40 challenged medium from cultured astrocytes	Elevation of the expression of NEP, an enzyme that degrades Aβ	[200]
	PUFAs oral administration in fat-1 transgenic mice and AQP4 knockout mice	PUFAs promoted Aβ clearance in fat-1 transgenic mice, but not in AQP4 knockout mice. PUFAs protected from Aβ-induced AQP4 polarization	[201]
*Complement cascade*	Genetic ablation of C3 gene in APP/PS1 mice	Reduction of glia at plaques	[202]
*Phagocytosis*	Fingolimod oral administration in APP/PS1 mice infected by *B. pertussis*	Increase in astrocyte phagocytosis of Aβ; reduction of GFAP immunoreactivity	[203]
	In vitro and in vivo downregulation of the Aβ-induced inflammasome, respectively in astrocytes and in 5xFAD mice	In vitro Aβ phagocytosis increase due to the release of the chemokine CCL3 and improved memory in vivo	[204]
*Cell reactivity*	Iron chelators deferoxamine and deferiprone treatment in Aβ-challenged astrocytes	Inhibition of Aβ-induced Lcn2	[205]
	Glu-DAPPD chronic administration in APP/PS1 mice	Reduction of Aβ aggregates as well as GFAP and Iba1 immunostaining. Cognitive functions improvement	[206]
	Downregulation of the JAK2-STAT3 pathway in hippocampal astrocytes of transgenic APP mice	Reduction of Aβ deposits; mice spatial learning improvement; control of pro-inflammatory genes	[207]
	Downregulation of the JAK2-STAT3 pathway in hippocampal astrocytes of transgenic 3xTg-AD mice	Full reversal of early synaptic and LTP alterations; short-term memory and reduced anxiety behavior improvements	[176,207]
	In vitro treatment with PEA of Aβ42-challenged primary astrocytes and mixed astrocytes-neurons cultures	Prevention of Aβ-induced neuronal loss and reduction of neuronal viability	[208]
	In vitro treatment with PEA of Aβ42-challenged mixed astrocytes-neurons cultures isolated from 3xTg-AD mice	Prevention of Aβ-induced neuronal loss and reduction of neuronal viability	[209]
	*In vitro* treatment with PEA of primary cortical astrocytes and mixed astrocytes-neurons cultures isolated from 3xTg-AD mice	Reduction of astrogliosis and improvement of neuronal viability	[210]
	Um-PEA treatment in glioma and neuroblastoma cells challenged by lipopolisaccaride and interferon γ	Improvement of cell viability; reduction of protein expression of both iNOS and COX-2	[211]
	Co-ultra PEALut administration for 14 days starting from the day that rats received a single intrahippocampal Aβ42 infusion	Prevention of Aβ-induced astrocyte hypertrophy, neuroinflammation; and BDNF and GDNF mRNA downregulation	[35]
*Oxidative stress*	Electromagnetic fields exposure of human and rat primary astrocytes challenged with Aβ or H_2_O_2_	Reduction of both ROS production and NADPH oxidase activity	[212]
	In vivo pelargonidin administration in rats subjected to an intrahippocampal injection of Aβ(25–35)	Raise in acetylcholinesterase and catalase activities. Improvement in cognitive performance	[213]
	In vivo treatment of *C. elegans* with monascin	Reduction of Aβ-toxicity and activation of the expression of several anti-oxidative genes	[214]
*Channels and receptors*	*In vivo* genetic ablation of the Ca^2+^-activated K^+^-channel KCa3.1	Improvements in memory performance and insulin signaling.Reduction of glial hypertrophy and tau hyperphosphorylation	[215]
	Chronic intracerebroventricular infusion of P2Y1R inhibitors in APP/PS1 mice	Reversal of structural and functional markers of astrocyte activation. Memory performance improvement	[216]
	Inhibition of adenosine recycle by J4 hippocampal infusion in APP/PS1 mice	Improvement of memory deficits	[217]
	Oral administration of istradefylline, an A2A antagonist, to transgenic APP mice	Memory improvements	[218]
*Astrocyte modulation*	Chronic um-PEA administration to 6-month-old 3xTg-AD mice	Reduction of cortical astrocyte hypertrophy and reactivity. Reduction in both cortical and hippocampal inflammation	[210,219]
	Chronic um-PEA administration to 12-month-old 3xTg-AD mice	Support for asthenic/atrophic astrocytes	[219]

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
