# Peer review of "Alternative Targets to Fight Alzheimer’s Disease: Focus on Astrocytes"

_biomolecules, 2021, doi:10.3390/biom11040600_

Round 1

Reviewer 1 Report

In this review paper, the authors summarize the main pathological mechanisms involved in Alzheimer´s disease (AD) with a special focus on glial cells and astrocytes. They then discuss and give an overview of most recent literature findings about therapeutic approaches targeting various astrocyte properties. This review is well written and nicely structured and highlights main recent findings. In summary, I support publication after minor spell checks have been done.

Author Response

In this review paper, the authors summarize the main pathological mechanisms involved in Alzheimer´s disease (AD) with a special focus on glial cells and astrocytes. They then discuss and give an overview of most recent literature findings about therapeutic approaches targeting various astrocyte properties. This review is well written and nicely structured and highlights main recent findings. In summary, I support publication after minor spell checks have been done.

Our reply: We thank Reviewer #1 for the positive comments. We thoroughly edited the text to correct it for language errors and typos.

Reviewer 2 Report

In their review, Caterina Scuderi and the team revisited a range of pathobiology that could be potential targets for the development of AD therapy. In particular, they focused on abstract dysfunction as a potential target and palmitoylethanolamide as a viable lead that could modulate this pathobiology. Overall, the review is well-planned and exhaustive; however, authors are recommended to address the following issues:

  1. Excess GABA released by reactive astrocytes spills over synaptic cleft and causes excessive tonic inhibition in the dentate gyrus, resulting in the impairment of hippocampal memory in AD (PMID: 24923909). Minimizing excessive tonic inhibition could therefore be another potential strategy against AD pathobiology. Authors should address this issue.
  2. In the section ‘Targeting astrocyte glutamate transporters’, authors are recommended to consult with the review (PMID: 26096891) for potential leads as Transcription/translational modulators of GLT-1.

Author Response

  1. <<Excess GABA released by reactive astrocytes spills over synaptic cleft and causes excessive tonic inhibition in the dentate gyrus, resulting in the impairment of hippocampal memory in AD (PMID: 24923909). Minimizing excessive tonic inhibition could therefore be another potential strategy against AD pathobiology. Authors should address this issue>>

Our reply: We have now discussed the topic suggested, citing the recommended article and some other related literature reports in the main text. Please see changes made between lines 440-455.

Since data included in Table 1 are restricted to papers published in the last 5 years, we could not include Wu et al. 2014 in the table.

  1. <<In the section ‘Targeting astrocyte glutamate transporters’, authors are recommended to consult with the review (PMID: 26096891) for potential leads as Transcription/translational modulators of GLT-1>>

Our reply: We included a sentence citing the review article suggested. Please see changes at lines 455-456.

Reviewer 3 Report

1. This manuscript is a well prepared text but still need a complete proofreading in English grammar and some spelling. 
2. The section 2 'Old and new pathophysiological mechanisms in AD' is redundant and should be simplified as most of the hypothesis and theories had be well summarized in other reviews.
3. The author focused on astrocyte function in AD but should not debase the role of activated microglia. Actually the activated microglia initiates the pro-inflammatory activation of other CNS cells in AD. The author should emphasize the important role of activated microglia and also mention the limitation of astrocyte target therapy.
4. The reactive astrocyte is the key of astroglia mediated neuroinflammation. Recently single nucleus RNAseq analysis identify some subgroups and markers of reactive astrocyte. Though this review focused on therapy, the author should not ignore these new essential information about reactive astrocyte and neuroinflammation.
5. For amyloid beta clearance, the author should not ignore astrocyte lysosome function. Because several astrocyte lysosomal genes had been identified as reactive astrocyte markers in newly released transcriptomics data.

Author Response

  1. <<This manuscript is a well prepared text but still need a complete proofreading in English grammar and some spelling.>>

Our reply: We apologize for the presence of some language errors. The manuscript has been revised by a native English speaker.

  1. <<The section 2 'Old and new pathophysiological mechanisms in AD' is redundant and should be simplified as most of the hypothesis and theories had be well summarized in other reviews.>>

Our reply: According to Reviewer#3 critique, we cut some sentences and shortened Section 2, with special regards to the longest parts discussing the amyloid cascade and the neurofibrillary tangles. Please, see text modifications on pages 4-6.

  1. <<The author focused on astrocyte function in AD but should not debase the role of activated microglia. Actually the activated microglia initiates the pro-inflammatory activation of other CNS cells in AD. The author should emphasize the important role of activated microglia and also mention the limitation of astrocyte target therapy.>>

Our reply: We agree with the Reviewer on the crucial role exerted by microglia in AD. Our intention was not to downplay it. However, we could not extensively discuss microglia activation, whose literature is massive, because our goal is to produce a review focused on astrocytes. Anyway, we have now included some sentences emphasizing the role of activated microglia in AD and highlighted the limitation of astrocyte target therapy. Please, see lines 287-289, 344-349, and 398-406.

Moreover, to address the points raised by another Reviewer, we have included other evidence on the role of microglia in AD in the revised version of the manuscript. Please, see the new version of paragraphs 2.4 and 3.

  1. <<The reactive astrocyte is the key of astroglia mediated neuroinflammation. Recently single nucleus RNAseq analysis identify some subgroups and markers of reactive astrocyte. Though this review focused on therapy, the author should not ignore these new essential information about reactive astrocyte and neuroinflammation>>

Our reply: We thank the Reviewer for this suggestion. In agreement, we added some recent evidence at lines 287-289, 332-333, and 394-396.

  1. <<For amyloid beta clearance, the author should not ignore astrocyte lysosome function. Because several astrocyte lysosomal genes had been identified as reactive astrocyte markers in newly released transcriptomics data.>>

Our reply: In agreement with the Reviewer's suggestion, we expanded the discussion at lines 576-579.

Reviewer 4 Report

Marta Valenza et al summary a review titled Alternative targets to fight Alzheimer’s disease: focus on astrocytes. The author focus on a specific approach that is targeting astrocytes. In conclusion PEA should be Phosphoprotein enriched in astrocytes (PEA)-15, which a potential therapeutic target in multiple disease states.
Yes, as the author mentioned, Microglia are the main immunocompetent cells of the nervous system and have non-neural origin; thus, they fulfill primarily defensive functions. In a recent publication in Cell, Ada Ndola et al. (2020) identified a negative regulator of c/EBPβ in bone marrow-derived macrophages (BMDMs), a factor elevated in the brains of patients with Alzheimer's disease, is regulated post-translationally by COP1, an E3 ubiquitin ligase whose loss leads to microglial activation and neurotoxicity. I suggest the author cite this cell paper and also provide perspective vision on astrocytes and Alzheimer's disease.
The central dilemma that arises in designing a therapy directed to the brain is passing through the blood-brain barrier (BBB), which has the function of isolating and protecting neural tissue, controlling the entry of molecules, and therefore hindering delivery. About 7,000 drugs have been assessed in the Comprehensive Medical Chemistry database, with only 5% able to cross the BBB to enter the CNS.

What hint we could get from the research about microglia and Alzheimer's disease? How we apply it to astrocytes and Alzheimer's disease?

Author Response

  1. <<Yes, as the author mentioned, Microglia are the main immunocompetent cells of the nervous system and have non-neural origin; thus, they fulfill primarily defensive functions. In a recent publication in Cell, Ada Ndola et al. (2020) identified a negative regulator of c/EBPβ in bone marrow-derived macrophages (BMDMs), a factor elevated in the brains of patients with Alzheimer's disease, is regulated post-translationally by COP1, an E3 ubiquitin ligase whose loss leads to microglial activation and neurotoxicity. I suggest the author cite this cell paper and also provide perspective vision on astrocytes and Alzheimer's disease.>>

Our reply: We thank the Reviewer for the comment. In agreement, we have included the importance of transcription factors involved in neuroinflammation citing the suggested paper. Please, see modifications at lines 251-259 and 295-299.

  1. << The central dilemma that arises in designing a therapy directed to the brain is passing through the blood-brain barrier (BBB), which has the function of isolating and protecting neural tissue, controlling the entry of molecules, and therefore hindering delivery. About 7,000 drugs have been assessed in the Comprehensive Medical Chemistry database, with only 5% able to cross the BBB to enter the CNS.>>

Our reply: We totally agree with the Reviewer. So, we have added this important consideration at lines 403-406.

  1. <<What hint we could get from the research about microglia and Alzheimer's disease? <<How we apply it to astrocytes and Alzheimer's disease?>>

Our reply: Microglia and astrocytes are two different cell types, with different functions and capabilities. They are both crucial for the correct functioning of the CNS. As such, results obtained in a cell type might not translate or apply to the other. Such a discussion would require extending and discussing a huge amount of research studies on microglia, which goes beyond the scope of the present review. Anyway, following Reviewer hints, we added some sentences underlining the crosstalk between different glial cell types. Please see the sentences at lines 296-299, 381-383, 401-403.

Round 2

Reviewer 3 Report

The manuscript was improved significantly and can be accepted for publication under current status.